# Effects of Aphicides on Pecan Aphids and Their Parasitoids in Pecan Orchards

**DOI:** 10.3390/insects12030241

**Published:** 2021-03-12

**Authors:** Eddie K. Slusher, Ted Cottrell, Angelita L. Acebes-Doria

**Affiliations:** 1Department of Entomology, University of Georgia, Tifton, GA 31793, USA; aacebes@uga.edu; 2USDA Southeastern Fruit and Tree Nut Research Laboratory, Byron, GA 31008, USA; ted.cottrell@usda.gov

**Keywords:** carbine, closer, insecticide resistance, parasitism, sefina

## Abstract

**Simple Summary:**

Insecticide application is the primary method for aphid management in commercial pecan orchards in the U.S. However, over-reliance and non-judicious insecticide use has led to numerous downsides, including insecticide resistance and impairment of beneficial insects. It is important to assess the efficacy and potential non-target impact of insecticides in order to create sustainable management programs. The objective of this study was to assess three insecticides (flonicamid, sulfoxaflor, and afidopyropen) on pecan aphids and their parasitoid 7, 14, and 21 days post-application in 2019 and 2020. In 2019, non-treated trees had up to 9-fold more aphids than treated trees 7 days post-application, but these differences diminished by 14 and 21 days after treatment application. Although aphid numbers were low during 2020, non-treated trees had more aphids in the lower canopy than most treated trees 7 days post-application. These differences diminished for the later assessments. Surprisingly, there was no significant difference in abundance of parasitoid adults or mummies between non-treated trees and treated trees. The results of this study indicate that growers have multiple products for aphid management, thus allowing product rotation to slow development of aphid resistance.

**Abstract:**

Aphids are important pests of pecans. Traditionally, insecticides have been the primary method of management. However, over-reliance and non-judicious use has led to resistance and damage to natural enemy populations. Therefore, frequent assessment of insecticides is necessary in order to monitor resistance development and non-target impacts. Aphicides, flonicamid, sulfoxaflor, and afidopyropen were assessed for their effects on pecan aphids and parasitoid, *Aphelinus perpallidus,* in a mature pecan orchard in 2019 and 2020. Post-application assessments were performed 7, 14, and 21 days post-application. Leaf samples from non-treated trees had greater aphid numbers than treated trees 7 days post-application with differences diminishing throughout the other two treatment periods in 2019. In 2020, aphid numbers were lower but leaf samples from non-treated trees had more aphids than treated trees 7 days post-application in the lower canopy. These differences again diminished 14 and 21 days post-application. There was no difference among treatments in number of parasitoid adults or mummies. These findings indicate that pecan growers have multiple potential options available for aphid management that do not negatively impact the primary pecan aphid parasitoid. Implications of the results on pecan aphid management are discussed.

## 1. Introduction

Pecan, *Carya illinoensis* (Wangenh.) K. Koch, (Fagales: Juglandaceae) is a native nut crop and ranks as one of the ten most important agricultural commodities in Georgia. In 2019, Georgia produced 52,204.4 bearing hectares with an average yield of 599.4 kg per hectare [1]. In 2019, pecan production in the USA amounted to USD 263,359,174 in farm-gate value [2]. As with most agricultural commodities, pecan is attacked by an assemblage of pests such as aphids (Hemiptera: Aphididae), mites (Acari: Tetranychidae), pecan weevil (Coleoptera: Curculionidae), hickory shuckworm (Lepidoptera: Tortricidae), pecan nut casebearer (Lepidoptera: Pyralidae), ambrosia beetles (Coleoptera: Curculionidae), and prionus rootborers (Coleoptera: Cerambycidae). The foliage, nuts, trunk, and roots of the tree are all targets of pest pressure [3]. The number of pests and the different parts of the tree being attacked cause issues for growers who rely on a wide variety of pest management tactics.

Aphids are serious pests whose feeding can compromise tree health in addition to negatively impacting nut quality and yield [3,4,5]. Three species of aphids feed on pecans: the yellow pecan aphid, *Monelliopsis pecanis* Bissell (Hemiptera: Aphididae), the blackmargined aphid, *Monellia caryella* (Fitch) (Hemiptera: Aphididae), and the black pecan aphid, *Melanocallis caryaefoliae* (Davis) (Hemiptera: Aphididae). Yellow pecan aphid and blackmargined aphid are collectively referred to as the “yellow aphid” complex. All three species feed on the leaves but cause different types of damage. Both the yellow aphid complex and black pecan aphid excrete honeydew as a by-product of feeding. The honeydew coats the leaf causing the leaf to turn glossy and sticky and, over time, leads to the development of sooty mold. Sooty mold-coated leaves have lower photosynthetic capabilities. In addition, black pecan aphid feeding elicits localized leaf chlorosis around the feeding area. Severe infestations of black pecan aphid can lead to defoliation of the tree. Insecticide application is the primary and often necessary method for aphid management in various systems. However, the reliance on insecticides, coupled with only a recent understanding of some of the resistance mechanisms present in many aphid species, has led to increased difficulty in managing aphids as insecticide resistance emerges across various systems [6,7,8,9,10,11]. Studies on green peach aphid (*Myzus persicae* Sulzer; Hemiptera: Aphididae) in the late 1990s found that carboxylesterases in the aphid’s body provided resistance to organophosphates, pyrethroids, and carbamates. This was only found years after the first case of aphicide resistance was reported in this species in 1955 [8]. Therefore, testing and assessment of numerous aphicides is important in order to evaluate effectiveness. In addition, evaluation of the non-target effects on beneficial insects is also critical. Non-judicious application of aphicides can destroy natural enemy populations such as parasitoids and predators leading to secondary outbreaks of other pests or a resurgence of the target pest [4,5,12,13,14].

One important organism that may be affected is *Aphelinus perpallidus* Gahan (Hymenoptera: Aphelinidae). This parasitoid attacks the yellow pecan aphid and the blackmargined aphid but not the black pecan aphid. They are known to parasitize both the nymphal and adult stages [14,15,16,17,18]. While little is known about their biology and life history, they are an important natural enemy of pecan aphid [15,16,17,18,19]. Their specialization on pecan aphids contrasts with most aphid predators, such as lacewings and ladybeetles, that are more generalist in nature and leave the host plant when prey is limited [20].

Another aspect of insecticide application in pecans that is poorly studied is the effects of insecticide application on pests and beneficial insects at different canopy heights. Mature pecan trees in orchards can vary in height from 4–20 m with exceptionally tall wild individuals exceeding 30 m [21]. Both previous research and unpublished research by the authors indicates that pest and predators vary in their usage of the upper and lower canopy of pecan trees [22]. In addition, previous research has shown that spray coverage decreases significantly as canopy height increases [23]. However, research that examines the effects of insecticide application on both pest and beneficial insects at different canopy heights is currently lacking.

The objective of this study was to assess the effects of flonicamid, sulfoxaflor, and two concentrations of afidopyropen. The aphicides used in this study can be used as rotational chemistries for aphid management because they have different modes of action per the Insecticide Resistance Action Committee (IRAC) [24]. Flonicamid is a group 29 insecticide that targets the chordotonal organ modulators causing the target species to cease feeding after 30 min, as well as reduces honeydew production [24]. Additional behavioral effects include light sensitivity and irregular, erratic movement [24]. The only evidence of resistance to flonicamid thus far has been found in the cotton aphid (*Aphis gossypii* Glover) attacking fruiting vegetables in Korea [25]. Sulfoxaflor is a group 4C insecticide targeting the nicotinic acetylcholine receptor competitive modulators. Green peach aphids treated with sulfoxaflor experience tremors and eventual paralysis. Due to its novel chemistry, there is little to no evidence of cross-resistance to sulfoxaflor [26]. Afidopyropen is a group 9D insecticide targeting the chordotonal organ transient receptor potential cation channel (TRPV) modulators. This action affects movement and feeding activity in the target pest leading to starvation [14].

We assessed the effects of these products on the three pecan aphid species, aphid mummies and adults of the primary pecan aphid parasitoid, *A. perpallidus*. In addition, we assessed the effects of these treatments on aphids and parasitoids in the upper and lower canopy of tall, mature pecan trees.

## 2. Materials and Methods

This study was conducted during the 2019 and 2020 growing seasons in Ty Ty, Georgia, USA on a 3.64 ha research orchard planted with mature “Desirable” pecan trees. All treatment applications were made using an airblast sprayer (CDP20P150P Air 32, Durand-Wayland, Inc., LaGrange, GA, USA) calibrated to deliver 935 L/ha. Trees were sprayed with fungicides every two weeks from May to August each year. In 2020, carbaryl (Carbaryl 4L^©^, 479.9 g a.i./liter, Drexel Chemical Company, Memphis, TN, USA) and zeta-cypermethrin (Mustang Maxx^©^, 95 g a.i./liter, FMC Corporation, Philadelphia, PA, USA) were applied at 4731.8 and 22.7 g a.i./ha, respectively, once a week for two weeks in order help increase aphid numbers for the study. Carbaryl and zeta-cypermethrin are members of the carbamate and pyrethroid insecticide families. Carbamates and pyrethroids are often not recommended for aphid management due to resistance problems and non-target effects on natural enemies. Spraying these insecticides often allows increases in aphid abundance due to decreases in natural enemies [5,27]. This was not carried out during the 2019 study when yellow and black aphids were more abundant. Our experimental design was a randomized complete block design consisting of four blocks. Treatments were randomly assigned to their own pair of pecan trees in each block. Pre-treatment aphid sampling was done on all pre-selected trees for the experiment on 9 September 2019 and 14 August 2020. On 10 September 2019 and 17 August 2020, each replicate pair within each block was treated with one of five treatments: flonicamid 207.01 mL/ha (Carbine^©^, 857.3 mL a.i/liter, FMC Corporation, Philadelphia, PA, USA), sulfoxaflor 203.3 mL/ha (Closer^©^, 3429.2 g a.i./liter, Corteva Agriscience, Wilmington, DE, USA), low rate afidopyropen 221.8 mL/ha (Sefina^©^, 720.12 g a.i./liter, BASF Ag Products, Research Triangle Park, NC, USA), high rate afidopyropen 443.6 mL/ha or a non-treated control. These rates are based on the recommended labelled rates. Treated trees were buffered from other treatments by being located in every other row and allowing at least two trees between each treated pair within the same row.

The average high temperature during the 2019 sampling period was 34.75 °C, and the average low temperature was 19.9 °C. The hottest day recorded during the sampling period was 38.5 °C on September 27th, while the coolest night was 16.1 °C on September 21st. Average rainfall during the sampling period was 0.06 mm per week. For 2020, the average high temperature was 32.78 °C, while the average low temperature was 22.4 °C. The hottest day during the sample period was 35.4 °C on August 18^th^ and the coolest night was 20.72 °C on September 7th. The average rainfall during the sampling period was 3 mm per week.

Post-spray assessments were carried out on 17 September (7 days), 25 September (14 days), and 2 October (21 days) during 2019, and 23 August (7 days), 31 August (14 days), and 7 September (21 days) during 2020. These days after treatment were selected based on the slow mortality rate of the mode of action of the selected insecticides [26].

For the assessment, five compound leaves were randomly selected from the upper (~6.1–9.1 m) and lower canopy (~1.5–1.8 m) of each treated and control tree. As the number of leaflets can vary, only the middle three pairs of leaflets in each leaf were sampled and taken to the lab. Leaflets were examined using a microscope (Luxeo 6z, Labomed^®^, Fremont, CA, USA) for “yellow aphid” nymphs and adults, black pecan aphid nymphs and adults and the mummies of parasitized aphids. In addition, one yellow sticky card (7.6 × 12.7 cm, Olson Products Inc.; Medina, OH, USA) was placed in the upper and lower pecan canopy (40 cards total) of one tree in each replicate pair on 17 (7–14 days) and 25 (14–21 days) September during 2019 and 23 (7–14 days) and 31 (14–21 days) August during 2020 for one week to assess populations of the aphid parasitoid *A. perpallidus,* in response to treatments.

During both years of the study, black pecan aphid was the dominant species averaging 94.15% and 60.70% across the sampling period in 2019 and 2020, respectively. In contrast to other multi-aphid species systems, previous studies of insecticidal effects on pecan aphid species were shown to be non-species specific [28,29]. Thus, all subsequent analyses of mean aphid populations included all aphid species. Data were examined for normality and homogeneity of variance and subjected to transformations (log + 1) when needed prior to analysis [30]. A two-way analysis of variance (ANOVA) with product and canopy location as fixed effects and block as a random effect was used to evaluate canopy location interactions, overall product effects, and canopy location effects. A one-way ANOVA was used to evaluate effects of aphicides in the upper and lower canopy separately. Tukey’s honestly significant difference (HSD) was used for post-hoc analysis at α = 0.05. All statistical analyses were performed using JMP^®^ Pro 14.1.0 (SAS Version 14.1.0, Cary, NC, USA).

## 3. Results

### 3.1. 2019 Results

#### 3.1.1. Effects of Insecticidal Treatment on Pecan Aphids

At 7 days post-application, there was no significant interaction between product and canopy location (Table 1). Aphid populations did not differ significantly between the upper (4.75 ± 2.05/leaf) and lower canopy (3.81 ± 6.98/leaf) (Table 1). An overall product effect was observed (Table 1). Aphid numbers were significantly different across the products in both canopies (Table 1). In the upper canopy, the non-treated control trees had a significantly greater number of aphids compared to the trees treated with aphicides (Table 1). In the lower canopy, non-treated control trees had a significantly greater number of aphids compared to the treated trees (Table 1). Trees sprayed with low-rate afidopyropen had significantly greater aphid populations compared to trees treated with sulfoxaflor (Table 1).

At 14 days post-application, there was no significant interaction detected between product and location (Table 1). No significant difference was detected between canopy locations (upper: 1.93 ± 0.33/leaf, lower: 2.28 ± 0.40/leaf Table 1). A significant difference in aphid numbers among the treatment groups was observed overall (Table 1). As well, significant differences among products were detected in the upper and lower canopies. In the upper canopy, trees sprayed with flonicamid had significantly greater numbers of aphids than trees treated with sulfoxaflor or high-rate afidopyropen (Table 1). In the lower canopy, control trees and trees treated with low-rate afidopyropen had significantly more aphids than trees treated with sulfoxaflor (Table 1).

At 21 days post-application, no significant interaction was found between product and location (Table 1). Significantly, more aphids were found in the lower canopy (4.35 ± 0.67/leaf) than in the upper canopy (1.65 ± 0.23/leaf) (Table 1). Overall, there was a significant difference in aphid numbers among treatment groups (Table 1). A significant difference in aphids among product treatments was found in both the upper and lower canopy (Table 1). In the upper canopy, significantly more aphids were found in the control than in sulfoxaflor (Table 1). In the lower canopy, significantly more aphids were found in the control trees than all other treated trees except for flonicamid (Table 1).

#### 3.1.2. Effects of Insecticidal Treatment on Aphid Mummies

There was no significant interaction between product and canopy location on aphid mummies across any sampling period (Table 2). Significantly more aphid mummies were found in the lower canopy than in the upper canopy at 14 days (upper: 0.045 ± 0.02//leaf, lower: 0.355 ± 0.13/leaf) and 21 days (upper: 0 ± 0/leaf, lower: 0.24 ± 0.04/leaf) post-treatment (Table 2). The aphicidal effects on the number of aphid mummies were not significantly different across any of the assessment periods (Table 2). No significant differences in aphicidal effects were found in either the upper or lower canopy (Table 2).

#### 3.1.3. Effects of Insecticidal Treatment on Adult Parasitoids

No interaction was found between product treatment and canopy location on the adult parasitoid numbers (Table 3). Significantly more adult parasitoids were found in the upper canopy (7.5 ± 2.66/card) than in the lower canopy (1.85 ± 0.67/card) at 14–21 days post treatment (Table 3). No significant difference in *A. perpallidus* populations among the insecticidal treatments was detected on either the 7–14 or 14–21 post-application intervals (Table 3). No significant difference in aphicidal effects on aphid numbers was found in either the upper canopy or lower canopy during either sampling period (Table 3).

### 3.2. 2020 Results

#### 3.2.1. 2020. Pre-Sample

While the pre-sample was only done in the lower canopy in 2019, the 2020 pre-sample was done in the both canopy locations. The pre-sample found significantly more aphids in the lower canopy (2.33 ± 0.17) than in the upper canopy (0.51 ± 0.17) (*p* < 0.0001). The same is true for mummies as well, who were significantly more abundant in the lower canopy (0.93 ± 0.11) than in the upper canopy (0.035 ± 0.03) (*p* < 0.0001).

#### 3.2.2. Effects of Insecticidal Treatment on Pecan Aphids

At 7 days post-assessment, no interaction was found between product treatment and canopy location (Table 4). Significantly more aphids were found in the lower canopy (0.665 ± 0.26/leaf) than in the upper canopy (0.125 ± 0.04/leaf). Overall aphid numbers were significantly different among treatments (Table 4). No significant differences in aphid numbers were found in the upper canopy among any of the treatment groups (Table 4). In the lower canopy, significantly more aphids were found in the control than in any of the treated trees except for sulfoxaflor (Table 4). Trees treated with sulfoxaflor had significantly more aphids than any of the other treated trees (Table 4).

During the 14-day post-assessment, no interaction was found between product treatment and canopy location (Table 4). Significantly more aphids were found in the lower canopy (0.73 ± 0.20/leaf) than in the upper canopy (0.17 ± 0.04/leaf). A significant difference among the treatments was found (Table 4). Aphid numbers across treatments in the upper canopy were similar (Table 4). In the lower canopy, significantly more aphids were found in the control than in sulfoxaflor (Table 4).

At 21 days post-assessment, no interaction was found between product treatment and canopy location (Table 4). Aphids were significantly more abundant in the lower canopy (2.18 ± 0.45/leaf) than in the upper canopy (0.50 ± 0.09/leaf) (Table 4). No significant overall production effect was found (Table 4). No significant effect was found at 21 days post-assessment among any of the treatment groups (Table 4). No significant effect of the aphicides on aphid numbers was found in either the upper or lower canopy (Table 4).

#### 3.2.3. Effects of Insecticidal Treatment on Aphid Mummies

No significant interaction was detected between the product and canopy location on aphid mummies across any of the sampling days (Table 5). Mummies were significantly more abundant in the lower canopy compared to the upper canopy during the 7-day (upper: 0.03 ± 0.01/leaf, lower: 0.975 ± 0.10/leaf), 14-day (upper: 0.025 ± 0.08/leaf, lower: 0.71 ± 0.08/leaf), and 21-day (upper: 0 ± 0/leaf, lower: 0.73 ± 0.14/leaf) sampling periods (Table 5). There was no significant difference in mummy abundance among any of the treatment groups across any of the sampling days (Table 5). There was no significant difference in aphicide effects on mummy number in either the upper or lower canopy during any of the sampling periods (Table 5).

#### 3.2.4. Effects of Insecticidal Treatment on Adult Parasitoids

No significant interaction was detected between the product and canopy location on adult parasitoids across any of the sampling periods (Table 6). There were no significant differences in parasitoid numbers between the upper (183.3 ± 31.7/card) and lower canopy (156.1 ± 35.3/card) on the 7–14 day or the upper (21.6 ± 4.6/card) and lower canopy (21.5 ± 6.09/card) on 14–21 day (Table 6). A significant overall product effect was detected during the 14–21-day sampling period (Table 6). However, no significant effect of aphicides on parasitoids was found in either the upper or lower canopy (Table 6.)

## 4. Discussion

Insecticide use is often necessary to manage aphids feeding on pecan foliage because aphid predators often arrive later than their prey or leave before the aphids begin to overwinter [20]. Predators, such as lacewings, may have lifecycles that lag behind their potential prey by as much as one week [20]. We demonstrated through this study that pecan growers have insecticidal options for aphid management. In 2019, trees sprayed with aphicides had significantly lower aphid populations than the non-treated control 7 days post-treatment application. Though aphid reduction varied among the aphicides, our analysis suggests that each aphicide can be used to successfully manage pecan aphids. The difference in aphid numbers between the control and treatments diminishes at 14 and 21 days post-application. This diminished effect could also be attributed to seasonal changes in pecan aphid populations, which usually peak in July and August before decreasing in late September and October when they begin to overwinter [3]. This is evident by the decrease in mean aphid numbers in the non-treated control, as well as the treated trees. The 2020 spray trial was carried out earlier in the year to better understand the effects of the aphicides on the yellow aphid complex whose populations were dropping naturally during the 2019 study. However, aphid numbers were low during the growing season and thus any diminishing effects were not apparent.

The aphicides used in this study are formulated to inhibit the feeding of insects with piercing–sucking mouthparts, such as aphids, whiteflies, and psyllids [14,26]. This not only makes them effective for managing the target insect but also helps prevent damage to natural enemy populations. Non-target effects are common in the pecan orchard setting and can result in a secondary outbreak of additional pests due to natural enemy suppression [5]. An example of this in the pecan system is the use of carbaryl for pecan weevil management causing a surge in pecan aphid populations as a result of reduced natural enemies [12]. Laboratory studies on the effects of insecticides on various aphid predators, including lacewings, lady beetles, and the mummies of *A. perpallidus*, indicated that no insecticide may be safe for all species in an orchard. Individual insecticides tested have reported differing effects among the predators and parasitoids [4]. In Mexico, application of tebufenozide and chlorpyrifos for management of hickory shuckworm had adverse effects on lacewings and ladybeetles [31].

Our findings in this spray trial found no significant difference in adult parasitoids or mummies in treated trees compared to the non-treated control across both years of the study. The exception to this was overall product on adult parasitoids from 14–21-day post-treatment application in 2020. However, we did not see these significant differences in the 7–14 day period. Therefore, it seems likely that these aphicides can be used to manage aphids while not adversely affecting *A. perpallidus*. In previous studies, no major non-target effects on beneficial insects, such as *Apis mellifera* L. (Hymenoptera: Apidae), and *Harmonia axyridis* Pallas (Coleoptera: Coccinellidae), were found for flonicamid [11,24]. Sulfoxaflor has low toxicity to terrestrial invertebrates, beneficial insects, and earthworms [26]. However, a study conducted on *Trichogramma* (Hymenoptera: Trichogrammatidae) found that sulfoxaflor exposure had lethal effects and impaired their parasitism ability [13]. The non-target effects of afidopyropen are poorly understood, due to it being a new insecticide. A study conducted on green peach aphid and its predator, the two spotted ladybeetle (*Adalia bipunctata* L.) (Coleoptera: Coccinellidae), found a significant difference in green peach aphid numbers between pre- and post-treatment but found no significant difference in two spotted ladybeetle larvae under the same parameters [14]. A laboratory study with soybean aphid found low toxicity to convergent ladybeetle (*Hippodamia convergens* Guerin-Meneville) (Coleoptera:Coccinellidae) and moderate toxicity to *Aphelinus certus* Yasnosh (Hymenoptera: Aphelinidae) [32]. The results of our study are similar to these results. However, previous literature also highlights the importance of species-to-species assessments of non-target effects. Our study is potentially one of the first to analyze the effects of afidopyropen on natural enemies in a field setting. In addition, this is one of the few recent studies to assess the effects of aphicides on natural enemies in an orchard production system.

Numbers of parasitized aphids were not significantly different among treatment groups throughout most of the study. Given that adult parasitoids were not affected significantly by the treatments, it is likely they were able to continue to parasitize aphids. It has also been documented that the mummified aphid may offer some degree of protection to the developing parasitoid. However, this depends largely on a number of factors such as the penetrability of the insecticide, as well as how soon the adult parasitoid will emerge [33]. Laboratory studies on aphids parasitized by *A. perpallidus* found that only methomyl and carbaryl caused significant mortality, 57 and 51%, respectively. All other insecticides tested were not toxic [4]. Lethal and sub-lethal effects of many modern insecticides on parasitoids in the pupal stage could be a basis of future studies.

Rainfall may have also been a factor in our study. A previous study by Kaakeh and Dutcher found a significant reduction in aphid numbers collected post-rainfall compared to pre-rainfall [34]. This is in accordance with other studies in different systems [35,36]. This may account for the lack of a significant difference in the upper canopy in 2020 compared to 2019 as the rainfall was greater in 2020 (3 mm) compared to 2019 (0.06 mm). The study mentioned above was performed in a single field season and may not accurately depict long-term trends. A multi-season study could help paint a bigger picture of the effects of rainfall on pecan aphid populations.

For this study, we analyzed the differences in population of aphids, mummies, and parasitoids between the upper canopy and lower canopy. In addition, we also assessed the interaction between the insecticides and canopy location. This was carried out in order to assess potential differences that canopy height had on treatment effects. Canopy height likely impacts spray coverage because the sprayer may be unable to supply adequate coverage to the upper canopy. Spray coverage area on spray cards placed at different heights in mature pecan trees decreases as height increases [23]. In our study, no significant interactions were found between insecticidal treatments and canopy location; however, aphid, mummy and parasitoid numbers differed significantly between upper and lower canopies. Aphids were more abundant in the lower canopies in 2019 at 21 days, and during all three sampling periods in 2020. Mummies were more abundant in the lower canopy during all sampling days during both years of the study except for 7 days post-application in 2019. Adult parasitoid distribution varied between upper and lower canopy only, during the 14–21-day sampling period in 2019, when adult parasitoids were more abundant in the upper canopy. The significantly higher numbers of aphids in the lower canopy are interesting given that spray coverage should have been greater in the lower canopy. This could have been due to higher populations of aphids in the lower canopy to begin with. This was evident based off our pre-sample, where aphids and mummies were more abundant lower canopy than in the upper canopy. Even though we found little effect of canopy location on *A. perpallidus* adults, considering the canopy height is also important for understanding differences in natural enemy populations. For instance, the number of lady beetle species inhabiting the pecan canopy is affected by canopy height—lady beetles respond negatively, positively, or neutrally to changes in canopy height [22]. Canopy height is a poorly studied factor that may have significant effects on aphicide efficacy as well as varying effects on natural enemy populations. Future studies are needed to address the potential effects of canopy height on aphicide application.

## 5. Conclusions

The results of this study indicate that sulfoxaflor, flonicamid, and afidopyropen can reduce aphid numbers without causing significant damage to mummy and adult parasitoid numbers. Assessment of aphid numbers on leaf samples revealed that aphid numbers were up to nine times greater in the non-treated control compared to the next highest treatment of low-rate afidopyropen in 2019. The difference in non-treated trees versus treated trees was much lower in 2020 due to a lower starting population than during 2019, but non-treated trees were still significantly greater in the lower canopy compared to most of the treatments. Effects on mummies and adults were minimal in both years with populations being statistically similar. Aphids and aphid mummies tended to be more abundant in the lower canopy. Adult parasitoids were similar in the upper and lower canopy with the exception of the 14–21-day sampling period in 2019. What caused these trends is a subject of further study. The findings of this study can be used to form the basis of a sustainable management program that potentially integrates insecticide management with biological control in pecans. However, future studies should focus on insecticide effects on natural enemies not assessed in this experiment such as lacewings and lady beetles.

## Figures and Tables

**Table 1 insects-12-00241-t001:** Mean ± SEM of total pecan aphids per leaf across treatments and sample days in 2019. Differing letters in columns indicate a significant difference between treatments (*p <* 0.05 Tukey’s honestly significant difference (HSD). Canopy-specific product effects and *p*-values for product*canopy interaction, overall product effect, and canopy location across sampling days are also shown. Application rates for each aphicide are based on the standard label rate.

Product	7-Day Post-Treatment		14-Day Post-Treatment		21-Day Post-Treatment	
	Upper	Lower	Upper	Lower	Upper	Lower
Control	18.93 ± 8.85 a	13.30 ± 3.77 a	2.90 ± 1.11 ab	3.03 ± 0.79 a	2.53 ± 0.59 a	9.25 ± 2.11 a
Flonicamid	0.95 ± 0.40 b	1.28 ± 0.72 bc	2.90 ± 0.79 a	2.00 ± 0.66 ab	2.08 ± 0.54 ab	4.18 ± 1.11 ab
Sulfoxaflor	0.48 ± 0.17 b	0.28 ± 0.14 c	0.75 ± 0.10 b	0.73 ± 0.20 b	0.70 ± 0.30 b	1.73 ± 0.46 b
Low-Rate Afidopyropen	2.40 ± 1.21 b	3.58 ± 1.45 b	2.30 ± 0.59 ab	3.95 ± 1.42 a	2.08 ± 0.43 ab	3.10 ± 0.77 b
High-Rate Afidopyropen	0.98 ± 0.65 b	0.60 ± 0.22 bc	0.80 ± 0.25 b	1.70 ± 0.48 ab	0.85 ± 0.36 ab	3.48 ± 1.17 b
Canopy-specific Product Effects	*F*_df_ = 10.4_4_ *p* = <0.0001	*F*_df_ = 12.2_4_ *p* = <0.0001	*F*_df_ = 4.20_4_ *p* = 0.0076	*F*_df_ = 3.67_4_ *p* = 0.0144	*F*_df_ = 4.00_4_ *p* = 0.0097	*F*_df_ = 6.37_4_ *p* = 0.0007
Product*Canopy Location (*p*-Value)	0.9293		0.3111		0.2829	
Overall Product Effect (*p*-Value)	<0.0001		0.0002		<0.0001	
Canopy Location Effect (*p*-Value)	0.9928		0.4548		<0.0001	

**Table 2 insects-12-00241-t002:** Mean ± SEM of mummified aphids per leaf across treatments and sample days in 2019. Differing letters in columns indicate a significant difference between treatments (*p <* 0.05 Tukey’s HSD). Canopy-specific product effects and *p*-values for product*canopy interaction, overall product effect, and canopy location across sampling days are also shown. Application rates for each aphicide are based on the standard label rate.

Product	7-Day Post-Treatment		14-Day Post-Treatment		21-Day Post-Treatment	
	Upper	Lower	Upper	Lower	Upper	Lower
Control	0.40 ± 0.26 a	0.53 ± 0.39 a	0.08 ± 0.04 a	0.45 ± 0.27 a	0.00 ± 0.00 a	0.35 ± 0.11 a
Flonicamid	0.10 ± 0.08 a	0.33 ± 0.15 a	0.05 ± 0.03 a	0.98 ± 0.58 a	0.00 ± 0.00 a	0.28 ± 0.08 a
Sulfoxaflor	0.05 ± 0.03 a	0.10 ± 0.04 a	0.10 ± 0.10 a	0.13 ± 0.08 a	0.00 ± 0.00 a	0.23 ± 10 a
Low-Rate Afidopyropen	0.20 ± 0.08 a	0.13 ± 0.08 a	0.00 ± 0.00 a	0.10 ± 0.05 a	0.00 ± 0.00 a	0.13 ± 0.08 a
High-Rate Afidopyropen	0.03 ± 0.03 a	0.23 ± 0.12 a	0.00 ± 0.00 a	0.13 ± 0.08 a	0.00 ± 0.00 a	0.23 ± 0.08 a
Canopy-specific Product Effects	*F*_df_ = 1.75_4_ *p* = 0.1631	*F*_df_ = 0.78_4_ *p* = 0.5455	*F*_df_ = 0.90_4_ *p* = 0.4740	*F*_df_ = 1.92_4_ *p* = 0.1315	*F*_df_ = 0_4_ *p* = 1	*F*_df_ = 0.82_4_ *p* = 0.5225
Product*Canopy Location (*p*-Value)	0.7333		0.2145		0.4973	
Overall Product Effect (*p*-Value)	0.2797		0.0988		0.4973	
Canopy Location Effect (*p*-Value)	0.2714		0.0038		<.0001	

**Table 3 insects-12-00241-t003:** Mean ± SEM of adult parasitoids per card across treatments and sample days in 2019. Differing letters in columns indicate a significant difference between treatments (*p <* 0.05 Tukey’s HSD). Canopy-specific product effects and *p*-values for product*canopy interaction, overall product effect, and canopy location across sampling days are also shown. Application rates for each aphicide are based on the standard label rate.

Product	7-14 Day Post-Treatment		14-21 Day Post-Treatment	
	Upper	Lower	Upper	Lower
Control	38.00 ± 34.02 a	43.75 ± 43.41 a	8.25 ± 4.44 a	0.75 ± 0.48 a
Flonicamid	7.75 ± 2.53 a	17.25 ± 9.56 a	2.25 ± 1.11 a	1.50 ± 0.65 a
Sulfoxaflor	16.50 ± 8.09 a	3.50 ± 2.18 a	13.25 ± 9.39 a	1.50 ± 0.96 a
Low-Rate Afidopyropen	2.50 ± 1.66 a	1.50 ± 1.19 a	8.25 ± 5.98 a	4.75 ± 2.87 a
High-Rate Afidopyropen	9.75 ± 8.09 a	7.75 ± 4.19 a	5.5 ± 3.11 a	0.75 ± 0.75 a
Canopy-specific Product Effects	*F*_df_ = 1.37_4_ *p* = 0.3011	*F*_df_ = 0.91_4_ *p* = 0.488	*F*_df_ = 0.35_4_ *p* = 0.8421	*F*_df_ = 1.70_4_ *p* = 0.2136
Product*Canopy Location (*p*-Value)	0.7221		0.5586	
Overall Product Effect (*p*-Value)	0.1922		0.7975	
Canopy Location Effect (*p*-Value)	0.186		0.01	

**Table 4 insects-12-00241-t004:** Mean ± SEM of total pecan aphids per leaf across treatments and sample days in 2020. Differing letters in columns indicate a significant difference between treatments (*p <* 0.05 Tukey’s HSD). Canopy-specific product effects and *p*-values for product*canopy interaction, overall product effect, and canopy location across sampling days are also shown. Application rates for each aphicide are based on the standard label rate.

Product	7-Day Post-Treatment		14-Day Post-Treatment		21-Day Post-Treatment	
	Upper	Lower	Upper	Lower	Upper	Lower
Control	0.23 ± 0.10 a	1.83 ± 1.13 a	0.13 ± 0.07 a	1.60 ± 0.74 a	0.63 ± 0.17 a	3.08 ± 1.10 a
Flonicamid	0.03 ± 0.03 a	0.15 ± 0.08 c	0.15 ± 0.07 a	0.55 ± 0.18 ab	0.53 ± 0.27 a	1.88 ± 0.75 a
Sulfoxaflor	0.28 ± 0.14 a	1.03 ± 0.47 ab	0.08 ± 0.06 a	0.13 ± 0.20 b	0.33 ± 0.16 a	1.98 ± 0.86 a
Low-Rate Afidopyropen	0.05 ± 0.03 a	0.15 ± 0.08 c	0.23 ± 0.10 a	0.45 ± 0.17 ab	0.43 ± 0.14 a	1.00 ± 0.29 a
High-Rate Afidopyropen	0.05 ± 0.05 a	0.18 ± 0.06 bc	0.28 ± 0.08 a	0.93 ± 0.26 ab	0.58 ± 0.26 a	2.95 ± 1.63 a
Canopy-specific Product Effects	*F*_df_ = 2.15_4_ *p* = 0.0973	*F*_df_ = 3.81_4_ *p* = 0.0121	*F*_df_ = 0.93_4_ *p* = 0.4567	*F*_df_ = 3.00_4_ *p* = 0.0327	*F*_df_ = 0.55_4_ *p* = 0.6988	*F*_df_ = 0.94_4_ *p* = 0.4612
Product*Canopy Location (*p*-Value)	0.1755		0.1072		0.7611	
Overall Product Effect (*p*-Value)	0.0007		0.0295		0.2755	
Canopy Location Effect (*p*-Value)	0.0013		0.0001		<0.0001	

**Table 5 insects-12-00241-t005:** Mean ± SEM of mummified aphids per leaf across treatments and sample days in 2020. Differing letters in columns indicate a significant difference between treatments (*p <* 0.05 Tukey’s HSD). Canopy-specific product effects and *p*-values for product*canopy interaction, overall product effect, and canopy location across sampling days are also shown. Application rates for each aphicide are based on the standard label rate.

Product	7-Day Post-Treatment		14-Day Post-Treatment		21-Day Post-Treatment	
	Upper	Lower	Upper	Lower	Upper	Lower
Control	0.00 ± 0.00 a	0.90 ± 0.13 a	0.00 ± 0.00 a	0.75 ± 0.11 a	0.00 ± 0.00 a	1.00 ± 0.27 a
Flonicamid	0.08 ± 0.05 a	0.75 ± 0.32 a	0.00 ± 0.00 a	0.73 ± 0.18 a	0.00 ± 0.00 a	0.63 ± 0.29 a
Sulfoxaflor	0.03 ± 0.03 a	1.20 ± 0.21 a	0.00 ± 0.00 a	0.70 ± 0.16 a	0.00 ± 0.00 a	0.70 ± 0.24 a
Low-Rate Afidopyropen	0.03 ± 0.03 a	1.18 ± 0.31 a	0.10 ± 0.08 a	0.60 ± 0.23 a	0.00 ± 0.00 a	0.60 ± 0.15 a
High-Rate Afidopyropen	0.03 ± 0.03 a	0.85 ± 0.12 a	0.03 ± 0.03 a	0.78 ± 0.27 a	0.00 ± 0.00 a	0.73 ± 0.23 a
Canopy-specific Product Effects	*F*_df_ = 0.74_4_ *p* = 0.5694	*F*_df_ = 0.99_4_ *p* = 0.4225	*F*_df_ = 1.46_4_ *p* = 0.2372	*F*_df_ = 0.21_4_ *p* = 0.9316	*F*_df_ = 0_4_ *p* = 1	*F*_df_ = 0.54_4_ *p* = 0.7086
Product*Canopy Location (*p*-Value)	0.3256		0.7088		0.7233	
Overall Product Effect (*p*-Value)	0.607		0.997		0.7233	
Canopy Location Effect (*p*-Value)	<0.0001		<0.0001		<0.0001	

**Table 6 insects-12-00241-t006:** Mean + SEM of adult parasitoids per card across treatments and sample days in 2020. Differing letters in columns indicate a significant difference between treatments (*p <* 0.05 Tukey’s HSD). Canopy-specific product effects and *p*-values for product*canopy interaction, overall product effect, and canopy location across sampling days are also shown. Application rates for each aphicide are based on the standard label rate.

Product	7-14 Day Post-Treatment		14-21 Day Post-Treatment	
	Upper	Lower	Upper	Lower
Control	219.00 ± 54.64 a	202.00 ± 68.87 a	45.00 ± 8.38 a	42.25 ± 18.40 a
Flonicamid	186.75 ± 75.21 a	92.75 ± 36.75 a	9.75 ± 2.56 a	14.25 ± 10.96 a
Sulfoxaflor	127.25 ± 52.16 a	144.25 ± 46.20 a	21.00 ± 11.25 a	38.25 ± 17.06 a
Low-Rate Afidopyropen	103.67 ± 12.12 a	200.50 ± 153.59 a	6.75 ± 1.80 a	3.75 ± 0.25 a
High-Rate Afidopyropen	261.25 ± 110.40 a	141 ± 69.23 a	25.5 ± 12.5 a	8.75 ± 3.47 a
Canopy-specific Product Effects	*F*_df_ = 0.30_4_ *p* = 0.8731	*F*_df_ = 0.34_4_ *p* = 0.8490	*F*_df_ = 1.89_4_ *p* = 0.1763	*F*_df_ = 2.76_4_ *p* = 0.0771
Product*Canopy Location (*p*-Value)	0.9714		0.844	
Overall Product Effect (*p*-Value)	0.6915		0.0064	
Canopy Location Effect (*p*-Value)	0.4042		0.4718	

## Data Availability

Non-Applicable.

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
