# Peer review of "Effects of Aphicides on Pecan Aphids and Their Parasitoids in Pecan Orchards"

_insects, 2021, doi:10.3390/insects12030241_

Round 1
Reviewer 1 Report
This manuscript investigates three insecticides, flonicamid, sulfoxaflor, and afidopyropen (4 treatments; low and high rate of afidopyropen) on pecan aphids and their parasitoids at 7, 14, and 21-days-post aphicides application in two years. The article is well written and clearly explained, but in my opinion, the experimental design has some limitations with no power to support conclusions. Two years but not synchronized in time and or with aphid picks; completely different aphid abundance; I also have some doubts about data analysis.
Few edits:
L90: I missed sections in material and methods.
L95-96: Maybe here, it would be important to explain briefly what carbaryl and zeta-cypermethrin are.
L98-99: And, here, give a brief explanation of how these substances help to increase aphid numbers!
L102-103: Pre-treatment sampling only in control treatment trees, why? why did not sample all treatments before spraying?
Any justification for this 1-month difference in the sampling period between years? Could be this the reason for the difference in aphid abundance?
L118-119: Please indicate…7, 14, and 21 days after aphicide treatments it is easier to follow!
L125: yellow sticky cards, How many?
L126-217: Why you did not sample from 0-7 days after treatment? For me, it would make sense to assess aphicide's effect.
L128: Add aphid parasitoid before A. perpallidus,
L129-130: Results section?
L133-137: Data analysis; Why you used 2 different analyses GLM-Poisson and ANOVA? Use only GLM-Poisson.
L136: post hoc analysis at α = 0.05; Tukey´s HSD?
L133-137: Maybe instead to cite [19] twice; I would report “The data were analyzed using the JMP statistical software [19].
L152: high decrease in aphid abundance in non-treated trees! Any justification?
L171: Very low natural parasitism, even in control but 7-day post-treatment tendency in control, high abundance at least 2 times higher
L171: Have you checked if the parasitoids were alive?
L177-1778: “Parasitized aphids...” Remove this sentence, it is stated in some way in the first sentence.
L181: put A. perpallidus in Italic.
L186: Table 3: Very big SEM, mainly in control? No differences (ANOVA) but the tendency, higher abundance in control.
L196 and L211: treatments
L214: More aphids and mummies in the lower canopy but more parasitoids in the upper canopy.
L215-218: Again, No differences (ANOVA) but there is a tendency…in control, there are cases of 2 to 5 times higher abundance than in some treatments.
Table 6 No differences?
L327-328: Yes, is true…but I wonder why? And even more if you state that is poorly studied.
L334: Reported in conclusion but not in results.
L336: parasitoid
L334-337: I am not sure, mainly in the case of adult parasitoid
L345: “...statistically similar in the upper...”, Not compared statistically
Author Response
Dear Reviewer,
Thank you for your review of our manuscript entitled “Effects of Aphicides on Pecan Aphids and Their Parasitoids in Pecan Orchards”. We have reviewed your helpful comments and suggestions and have made necessary changes based on your suggestions as well as those of your fellow reviewers. Provided below is our response to each of the comments and suggestions. Our responses are in blue text.
We hope you find these revisions to your satisfaction.
Comments and Suggestions for Authors
This manuscript investigates three insecticides, flonicamid, sulfoxaflor, and afidopyropen (4 treatments; low and high rate of afidopyropen) on pecan aphids and their parasitoids at 7, 14, and 21-days-post aphicides application in two years. The article is well written and clearly explained, but in my opinion, the experimental design has some limitations with no power to support conclusions. Two years but not synchronized in time and or with aphid picks; completely different aphid abundance; I also have some doubts about data analysis.
We appreciate your review of this manuscript. We have revised our statistical analysis. Below are our specific comments to your review. We noticed that the lines you have specified do not match up with the document we have, this might have been a formatting error. We have tried our best to acknowledge your comments.
Few edits:
L90: I missed sections in material and methods.
Sorry that happened, must have been a formatting error. If you need to see the missing sections, please let us know those missed and we can find a way to send them to you..
L95-96: Maybe here, it would be important to explain briefly what carbaryl and zeta-cypermethrin are.
Great suggestion! We have addressed this comment in the paper. Please see lines 123 – 127 in the revised manuscript.
L98-99: And, here, give a brief explanation of how these substances help to increase aphid numbers!
See previous comment.
L102-103: Pre-treatment sampling only in control treatment trees, why? why did not sample all treatments before spraying?
That was poor wording on our part, we did sample from all treatments before spraying. We have revised this to make it clearer (L 130 - 131).
Any justification for this 1-month difference in the sampling period between years? Could be this the reason for the difference in aphid abundance?
We sampled earlier in 2020 in order to better understand the effects of aphicides on yellow pecan aphids. Yellow pecan aphid numbers naturally drop in September and October and we want to better include them in the analyses. The reason for the difference in aphid abundance was likely not due to the time of the year as August is peak yellow aphid season. This was likely due to low pecan aphid population numbers during the 2020 growing season. This justification can be seen in the paper in lines 298 - 307.
L118-119: Please indicate…7, 14, and 21 days after aphicide treatments it is easier to follow!
We have included this suggestion in the narrative (lines 156 - 161).
L125: yellow sticky cards, How many?
Great suggestion, we have included this information in the narrative (line 158).
L126-217: Why you did not sample from 0-7 days after treatment? For me, it would make sense to assess aphicide's effect.
These days after treatment were selected based on label recommendations and the slow mortality rate of the mode of action of the selected insecticides (line 148 - 149).
L128: Add aphid parasitoid before A. perpallidus,
Done, please see line 159.
L129-130: Results section?
We were unable to figure out exactly what you were referring to due to the discrepancy in our line numbers. We may need further clarification to address this issue.
L133-137: Data analysis; Why you used 2 different analyses GLM-Poisson and ANOVA? Use only GLM-Poisson.
We have made this change. We decided to go with a 2-way ANOVA with product and canopy location as fixed effects and block as a random effect based on the recommendation of another reviewer. Assumptions of parametric tests were checked and transformations were made prior to analyses. Subsequently, ANOVA r-squared values were checked for model appropriateness.
L136: post hoc analysis at α = 0.05; Tukey´s HSD?
We have made changes to this section for clarification. Hope this helps.
L133-137: Maybe instead to cite [19] twice; I would report “The data were analyzed using the JMP statistical software [19].
We have made this change, please see lines 165 - 172.
L152: high decrease in aphid abundance in non-treated trees! Any justification?
Please see lines 298 - 307 for justification on why we believed this occurred.
L171: Very low natural parasitism, even in control but 7-day post-treatment tendency in control, high abundance at least 2 times higher
Thanks for the comment, this may be an interesting future study to look at. Maybe its due to movement of parasitoids into control trees from treated tree?
L171: Have you checked if the parasitoids were alive?
No, we didn’t. Adult parasitoids are usually dead on sticky cards and we did not keep mummies for observation in the lab to see if they survived to emergence. That may make an interesting future study though.
L177-1778: “Parasitized aphids...” Remove this sentence, it is stated in some way in the first sentence.
We have made this revision, hope that makes things a little clearer.
L181: put A. perpallidus in Italic.
Thanks for catching that, we have done this now.
L186: Table 3: Very big SEM, mainly in control? No differences (ANOVA) but the tendency, higher abundance in control.
We have re-ran the analysis with more model effects and have adjusted the tables and results as necessary, hope that helps.
L196 and L211: treatments
Thanks for catching those. We have made these corrections.
L214: More aphids and mummies in the lower canopy but more parasitoids in the upper canopy.
It is a very interesting phenomenon we have witnessed in other studies. We are not sure why they are distributed this way, but it is pretty interesting. May make a good future study.
L215-218: Again, no differences (ANOVA) but there is a tendency…in control, there are cases of 2 to 5 times higher abundance than in some treatments.
Table 6 No differences?
We have adjusted our analysis and added additional modeling effects in order to correct this.
L327-328: Yes, is true…but I wonder why? And even more if you state that is poorly studied.
It is a very interesting topic. I am actually looking at this more in depth in another study that focuses specifically on elevation. The effects of elevation on the distribution of organisms in the agricultural system has not been investigated much, and there is still a lot of factors that need to be looked into.
L334: Reported in conclusion but not in results.
Thanks for the comment, we are not sure which line you’re referring to due to a discrepancy in your line numbers and ours. Further clarification may be needed
L336: parasitoid
Thanks for catching that. We have made this change.
L334-337: I am not sure, mainly in the case of adult parasitoid
We have adjusted the statistical analysis, hope the helps to correct this concern.
L345: “...statistically similar in the upper...”, Not compared statistically
Thanks for catching that. We have made this change.
Reviewer 2 Report
This manuscript investigates the “Effects of Aphicides on Pecan Aphids and Their Parasitoids in Pecan Orchards”. The manuscript describes the insecticide effects on aphid species and parasitoid Aphelinus perpallidus. This is an interesting study; however, the manuscript needs a good revision before it is acceptable for publication. Please see my specific comments below:
L.16: …days after treatment application…
L.17: Delete “significantly”.
L.28: Delete “using a randomized complete block design”.
L.38: Keywords should be in alphabetic order.
L.45: … mites (Acari: Tetranychidae)…
L.78: Pull apart “4-20m” by “4-20 m”. Please, correct in all manuscript.
L.79: Again, pull apart “30m” by “30 m”.
Ls.85-86: In introduction section, provide the mode of action of each insecticide used in this study.
L.87: Also, provide a brief paragraph about importance and biological information of wasp parasitoid (Aphelinus perpallidus).
Ls.99-100: Which aphid species?
Ls.103-110: Explain why these doses rate (flonicamid 207.01 ml/ha, sulfoxaflor 203.3 ml/ha, low rate afidopyropen 221.8 ml/ha, and high rate afidopyropen 443.6 ml/ha) were used in bioassays.
Ls133-135: In particular, the GLM analyses of the number of life aphids and mummified should include aphid/leaf (upper/lower canopy) and treatment (insecticides), plus the interaction aphid/leaf x treatment as predictors.
Ls.135-136: But, which post hot analysis was carried out? Does this include a one-way ANOVA? Explain.
Tables 1, 2, 3, 4, 5, and 6 (see legends): …Different letters within each column signify a significant difference… Please, revise this sentence to correct grammatical errors and eliminate wordiness.
L.196: … the treatments was…
L.205: …low-rate afidopyropen (Table 4)…
L.211: …all treatments but…
Ls.225-237: In discussion section, this paragraph is irrelevant and not provides a further debate. Summarize of delete.
Ls.251-263: This information should be in introduction section.
L.264:…with piercing-sucking mouthparts…
L.211-318: Again, this paragraph deviates from the focus of your study. The rainfall effects were not investigated in this study.
L.336: …and adult parasitoid numbers…
Author Response
Dear Reviewer,
Thank you very much for your responses to my manuscript entitled “Effects of Aphicides on Pecan Aphids and Their Parasitoids in Pecan Orchards”. I and my co-authors have reviewed your helpful comments and suggestions and have made necessary changes based on your suggestions as well as those of your fellow reviewers. Provided below are our response to each of the comments and suggested. Our responses are in blue text.
We hope that you find these revisions to your satisfaction and that you find this revised manuscript suitable for publication in Insects.
This manuscript investigates the “Effects of Aphicides on Pecan Aphids and Their Parasitoids in Pecan Orchards”. The manuscript describes the insecticide effects on aphid species and parasitoid Aphelinus perpallidus. This is an interesting study; however, the manuscript needs a good revision before it is acceptable for publication. Please see my specific comments below:
Thank you for your comments, we have tried to address your comments to the best of our ability. Hope you are satisfied with what we have done.
L.16: …days after treatment application… Done
L.17: Delete “significantly”. Done
L.28: Delete “using a randomized complete block design”. Done
L.38: Keywords should be in alphabetic order. Thanks, we have corrected this issue.
L.45: … mites (Acari: Tetranychidae) Done
L.78: Pull apart “4-20m” by “4-20 m”. Please, correct in all manuscript.
L.79: Again, pull apart “30m” by “30 m”. Thanks for catching these, we have gone through and made revisions accordingly.
Ls.85-86: In introduction section, provide the mode of action of each insecticide used in this study. We moved the paragraph about insecticide modes of action to the introduction. Please see lines 94 – 108.
L.87: Also, provide a brief paragraph about importance and biological information of wasp parasitoid (Aphelinus perpallidus). Good suggestion! I have done this. Please see lines 78 - 84 for this information.
Ls.99-100: Which aphid species? I have made changes to this to include specific aphid names. Please see line 126-127.
Ls.103-110: Explain why these doses rate (flonicamid 207.01 ml/ha, sulfoxaflor 203.3 ml/ha, low rate afidopyropen 221.8 ml/ha, and high rate afidopyropen 443.6 ml/ha) were used in bioassays.
These rates were selected based upon the range of recommended application rates provided by the manufacturer on the label. Please see line 136.
Ls133-135: In particular, the GLM analyses of the number of life aphids and mummified should include aphid/leaf (upper/lower canopy) and treatment (insecticides), plus the interaction aphid/leaf x treatment as predictors. We have made this change. After checking data for assumptions of parametric tests and doing the necessary transformations, we decided to use a 2-way ANOVA with product and canopy location as fixed effects and block as a random effect. We have adjusted the tables and narrative, as necessary.
Ls.135-136: But, which post hot analysis was carried out? Does this include a one-way ANOVA? Explain. Sorry for the confusion we have adjusted the narrative to make this clearer to the reader. Please see lines 164 - 172.
Tables 1, 2, 3, 4, 5, and 6 (see legends): …Different letters within each column signify a significant difference… Please, revise this sentence to correct grammatical errors and eliminate wordiness. We have done our best to correct this issue, please see the legends on the new tables.
L.196: … the treatments was… Done
L.205: …low-rate afidopyropen (Table 4)… Done
L.211: …all treatments but… Done
Ls.225-237: In discussion section, this paragraph is irrelevant and not provides a further debate. Summarize of delete. We have summarized the paragraph and merged with the following one. Please see lines 290 - 306.
Ls.251-263: This information should be in introduction section. We have moved this, you will now see it in the introduction.
L.264:…with piercing-sucking mouthparts… Done
L.211-318: Again, this paragraph deviates from the focus of your study. The rainfall effects were not investigated in this study. True, but we did include weather data for both our sampling periods (Lines 139 -145).
L.336: …and adult parasitoid numbers… Done
Reviewer 3 Report
Materials and methods
100-110 It is not clear to me why sampling of aphids was performed on all experimental blocks before treatments? But only in control. Maybe I didn't understend?
Results
144-168 Aphids tend to be located on the higher parts on the canopy. What is the reason that the larger population on lower ones? On the treated part of the experiment it would be expected that the lower part was better treated than the higher. What is the reason for the reverse situation? You cite precipitation, is that the reason?
Discussion
Predators like lacewings and lady beetles are not the subject of this study, but parasitoids (Aphelinus perpallidus). These are no need to state in the discussion something that has not been researched.
Author Response
Dear Reviewer,
Thank you very much for your responses to my manuscript entitled “Effects of Aphicides on Pecan Aphids and Their Parasitoids in Pecan Orchards”. I and my co-authors have reviewed your helpful comments and suggestions and have made necessary changes based on your suggestions as well as those of your fellow reviewers. Provided below are our response to each of the comments and suggested. Our responses are in blue text.
We hope that you find these revisions to your satisfaction and that you find this revised manuscript suitable for publication in Insects.
Comments and Suggestions for Authors
Materials and methods
100-110 It is not clear to me why sampling of aphids was performed on all experimental blocks before treatments? But only in control. Maybe I didn't understend? Sorry about the confusion, we did perform pre-sampling on all treatments. I have adjusted the wording for clarity. Please see lines 129 - 130.
Results
144-168 Aphids tend to be located on the higher parts on the canopy. What is the reason that the larger population on lower ones? On the treated part of the experiment it would be expected that the lower part was better treated than the higher. What is the reason for the reverse situation? You cite precipitation, is that the reason? We are not entirely sure what the reason for this is. We have suggested rain as a possible factor. We have witnessed this in our other studies as well. We have also noticed the adult parasitoid prefer the upper canopy to the lower one. It has not been properly assessed as to what biotic and abiotic factors drive these organisms to prefer one location over the other in agricultural systems. It would make a very interesting future study.
Discussion
Predators like lacewings and lady beetles are not the subject of this study, but parasitoids (Aphelinus perpallidus). These are no need to state in the discussion something that has not been researched.
You make a good point. However, a good part of the literature on pecan predators and parasitoids focuses on these two organisms. A. perpallidus is poorly studied in the pecan system therefore, we found it appropriate to draw parallels between A. perpallidus and predators found in the same system.
Round 2
Reviewer 1 Report
The authors have accepted most of my suggestions. Suggestions from the other reviewer also improve the quality of the manuscript.